# Proliferation in Postmenopausal Endometrial Polyps—A Potential for Malignant Transformation [note 1]

**DOI:** 10.3390/medicina55090543

**Published:** 2019-08-28

**Authors:** Lina Adomaitienė, Rūta Nadišauskienė, Mahshid Nickkho-Amiry, Arvydas Čižauskas, Jolita Palubinskienė, Cathrine Holland, Mourad W Seif

**Affiliations:** 1Department of Obstetrics and Gynaecology, Medical Academy, Lithuanian University of Health Sciences, LT-44307 Kaunas, Lithuania; 2Department of Obstetrics and Gynaecology, University of Manchester and St. Mary’s Hospital, Manchester M13 9WL, UK; 3Department of Patological Anatomy, Medical Academy, Lithuanian University of Health Sciences, LT-44307 Kaunas, Lithuania; 4Department of Histology and Embryology, Medical Academy, Lithuanian University of Health Sciences, LT-44307 Kaunas, Lithuania

**Keywords:** endometrial polyps, postmenopausal, malignancy, proliferation, Ki-67

## Abstract

*Background and objectives:* Endometrial polyps in asymptomatic postmenopausal women are often incidentally found, yet only 1.51% of them are malignant. Their potential for malignant transformation has not been adequately addressed. The aim of this study was to investigate the proliferation within endometrial polyps as one of the indicators of their malignization potential in asymptomatic postmenopausal women. *Materials and Methods:* Immunohistochemical studies of Ki-67 were performed. Cases included 52 benign postmenopausal polyps, 19 endometrioid carcinoma with coexisting benign polyps, 12 polyps with foci of carcinoma and 4 cases of polyps, which later developed carcinoma. The control group included 31 atrophic endometria and 32 benign premenopausal polyps. Ki-67 was scored in either 10 or 20 “hot spot” fields, as percentage of positively stained cells. *Results*: The median epithelial Ki-67 score in postmenopausal benign polyps (4.7%) was significantly higher than in atrophic endometria (2.41%, *p* < 0.0001) and significantly lower than in premenopausal benign polyps (11.4%, *p* = 0.003) and endometrial cancer (8.3%, *p* < 0.0001). Where endometrial polyps were found in association with endometrial carcinoma, Ki-67 was significantly higher in cancer (*p* < 0.0001). No significant difference was found between Ki-67 scores of cancer focus and of the polyps tissue itself, respectively 2.8% and 4.55%, *p* = 0.37. Ki-67 expression, where polyps were resected and women later developed cancer, was not significantly different (*p* = 0.199). *Conclusion*: Polyps from asymptomatic postmenopausal women showed significantly more proliferation in both epithelial and stromal components than inactive atrophic endometria but less than premenopausal benign polyps and/or endometrial cancer. Benign postmenopausal endometrial polyps exhibit low proliferative activity, suggesting low malignant potential and may not require resection in asymptomatic women.

## 1. Introduction

Endometrial polyps (EPs) are outgrowths of endometrial tissue and are composed of varying amounts of glands and fibrotic stroma containing thick-walled blood vessels covered by epithelium [1]. The prevalence of EPs in the general population is approximately 8%, affecting up to 20% of postmenopausal women [2,3,4]. The majority of postmenopausal EPs are benign but a small proportion, that is, 4.47% among symptomatic postmenopausal women in comparison to 1.51% of asymptomatic women with polyps, are malignant [5].

Improved performances in gynaecological ultrasonography have enabled an increasing number of often asymptomatic EPs to be detected [6]. Most of these polyps are removed surgically, as a precautionary measure in order to avoid missing a case of endometrial cancer (EC) [6]. Hysteroscopic polypectomy is the gold standard to treat EPs and to obtain specimens for histological evaluation. However, there is still a continuing debate regarding when to offer hysteroscopic polypectomy and how to best manage asymptomatic postmenopausal women with incidentally identified EPs. In this situation, gynaecologists must decide whether to prioritize the risk of malignancy associated with EPs versus the health care costs and complications related with invasive procedures [7].

Carcinogenesis is a multistep process that involves the induction of mutational activation in tumour suppressor genes, increased cellular proliferation and angiogenesis for tumour growth [8]. Immunohistochemical methods are useful for detecting biomarkers of possible prognostic importance for a number of cancer types. Among the proteins that are responsible for cellular proliferation, Ki-67 is used as a cellular proliferation marker and might be used as a diagnostic and prognostic tool in EC. A number of studies describe Ki-67 as a prognostic biomarker, showing correlation between Ki-67 score and known pathological prognostic variables of EC, including grade, stage, depth of myometrial invasion and cancer outcomes [9,10,11,12,13].

The present study aimed to investigate proliferative activity in postmenopausal endometrial polyps as one of the indicators of their malignization potential in asymptomatic women. 

## 2. Materials and Methods

### 2.1. Ethics

Kaunas Regional Biomedical Research Ethics Committee, Lithuanian University of Health Sciences, Lithuania, provided a consent waiver and approved the study (BE-2-14, 17 February 2016). All samples were coded using unique identifiers.

### 2.2. Design

A retrospective study was conducted at Lithuanian University of Health Sciences, Kaunas, Lithuania.

### 2.3. Patients and Tissue Collection

The selected cases were obtained from the Pathology Department at the Hospital of Lithuanian University of Health Sciences and included 52 benign postmenopausal endometrial polyps with no history of bleeding, 19 endometrial carcinomas which had coexisting benign endometrial polyps, 12 endometrial polyps with foci of endometrial carcinoma and four patients with polyps who later developed endometrial carcinoma. The controls included 31 cases of atrophic endometrium and 32 premenopausal benign endometrial polyps. Women were considered postmenopausal if they reported a period of amenorrhea of at least 12 months duration after an age of 45 years.

Endometrial samples were submitted either as endometrial curetting, polypectomy or hysterectomy specimens of the patients diagnosed with endometrial polyps, endometrial carcinoma or endometrial atrophy. Two independent expert pathologists reviewed haematoxylin-eosin sections in order to confirm the histologic diagnosis. In the endometrial carcinoma cases stage was determined by The International Federation of Gynaecology and Obstetrics (FIGO) surgical staging, 2009 [14].

### 2.4. Immunohistochemistry

Histological sections of 4 µm were obtained. Immunohistochemistry was performed on paraffin sections using the avidin-biotin-peroxidase complex method for the identification of the antigen binding sites. Monoclonal mouse antihuman Ki-67 antibody (Anti-Ki-67, clone MIB-1; Dako Corp., Glostrup, Denmark) was used for the identification of the Ki-67 nuclear protein. Sections were heated at 62 °C for 1 h, de-paraffinized in xylene and rehydrated in graded alcohol, then boiled in buffer (pH 9.0) for 3 min at 120 °C. After endogenous peroxidase has been blocked, they were incubated with the pre-diluted primary antibody MIB-1 (Dako Corp., Glostrup, Denmark) with 1:100 dilutions. After 30 min incubation the slides were treated with anti-mouse IgG as the secondary antibody. Reactions were visualized using the avidin-biotin-peroxidase complex with 3,3′ diaminobenzidine (DAB, Sigma Chemical Co., St. Louis, MO, United States) as a chromogen. Counterstaining was performed with Mayer’s haematoxylin. Finally, the sections were dehydrated and mounted. 

For each batch of slides, when staining with the primary antibody, negative and positive controls were used according to the manufacturer’s recommendation.

### 2.5. Evaluation of Immunohistochemical Staining

All slides were examined under light microscopy. Image capture and analysis was performed using an Olympus BX 53F (Tokyo, Japan) light microscope and the digital camera QImaging EXi AQUA (Surrey, CO, Canada). Digital image analysis was performed using the image analysis software Image-Pro Plus (version 7.0). 

We used comprehensive recommendations on pre-analytical and analytical assessment and interpretation and scoring of Ki-67 staining that were formulated by the International Ki-67 in Breast Cancer Working Group in 2011 [15]. 

The quantitative evaluation of immunostaining for Ki-67 was performed by assessing the extent of nuclear positivity in epithelial and stromal compartments separately. The area with the highest level of nuclear staining was selected (hot-spot counting), the number of positively stained nuclei in at least 1000 glandular cells was assessed under 40× magnification and results were expressed as percentage staining and referred to as Ki-67 score. In each case, a number of 10 (in cases of one type of tissue—atrophic endometrium; benign EPs) or 20 fields (in cases of two different types of tissues—endometrial carcinoma which had coexisting benign EP; EPs with foci of endometrial carcinoma) were counted and averages were then calculated. 

### 2.6. Statistical Analysis

The necessary number of patients to be included in the study was estimated by differences in means using the sample size software Epi Info^TM^ 3.0, assuming an alpha error of 0.05 and a statistical power of 80%.

Calculations were carried out using Statistical Package of Social Science, Windows version 23 (Statistical Package for the Social Sciences (SPSS), IBM, Brøndby, Denmark). The Shapiro–Wilk normality test was used to examine the compatibility between the measured variables and the normal distribution. As data were not distributed normally, pairwise comparisons were performed by the Mann−Whitney *U* test. The Kruskal−Wallis test was used to examine the correlation of Ki-67 expression in relation to tumour grade and stage, followed by a post hoc test. The numerical data are presented with median and interquartile ranges and the results were considered significant with a *p* value of less than 0.05.

## 3. Results

### 3.1. Demographics

The median age of the patients diagnosed with malignant polyps was 63.5 (range—53–71) years, for the atrophic endometrium patients, it was 67.8 (54–88); for the benign premenopausal polyps patients, it was 41.2 (27–51); and for the benign postmenopausal polyps patients, it was 66.9 (53–89).

### 3.2. Histopathology

The expression of Ki-67 was examined in 150 patients—52 benign postmenopausal endometrial polyps with no bleeding history, 19 endometrial carcinoma which had coexisting benign endometrial polyps, 12 endometrial polyps with foci of endometrial carcinoma and four patients with polyps who later developed endometrial carcinoma (the latter analysed as four postmenopausal endometrial polyps and four endometrial carcinoma cases), 31 atrophic endometria and 32 premenopausal benign endometrial polyps (Figure 1).

The histologic type of carcinoma was endometrioid adenocarcinoma in all cases. Polyps with malignant features (*n* = 12) showed no myometrial invasion in more than half (58.3%) of cases, as EC was confined to a polyp in the same amount of the patients. No patients were diagnosed with the stage IB in the latter group. Seven out of 12 cases had a high degree of histodifferentation (G1) and only one case had poorly differentiated (G3) carcinoma (Table 1). None of the patients had lymphovascular space involvement or lymph node metastasis.

### 3.3. Ki-67 Score—Between Group Analysis

The median epithelial and stromal Ki-67 scores in postmenopausal benign polyps (4.67 and 0.045%) were significantly higher in both compartments than in atrophic endometrium (2.41 and 0.01%, *p* < 0.0001) and significantly lower than in premenopausal benign polyps (11.4 and 0.12%, *p* = 0.003) and endometrial cancer (8.3 and 0.43%, *p* < 0.0001) (Table 2; Figure 2). Out of the cases of atrophic endometrium, 22.6% showed completely negative staining of Ki-67.

Where EPs were found in association with EC, epithelial and stromal Ki-67 median scores were significantly higher in cancer tissue, than in polyp, at 33.5 and 1.08% versus 4.4 and 0.03%; *p* < 0.0001. The group of EPs with foci of EC did not reveal a statistically significant difference between the polyp tissue itself and the EC focus, with overall low scores of Ki-67 in epithelial and stromal components, at 4.55 and 0.05% versus 2.8 and 0.18%, respectively; *p* = 0.37. The median period from benign postmenopausal polyp diagnosis till EC development was 4.25 years. In two cases, EC was diagnosed 7 years after polyp resection. Ki-67 expression in epithelial and stromal compartments of earlier resected polyps (1.3 and 0.02%) did not differ significantly from analogous scores (1.9 and 0.28%) in tissues of EC, which developed later in these patients (*p* = 0.199) (Table 2; Figure 3).

The median epithelial Ki-67 scores of G1, G2 and G3 of endometrioid adenocarcinoma were 6.15%, 7.10%, 13.00% (Figure 4b and Figure 5) and of stages IA and IB were 8.30% and 9.15%, respectively (Figure 4a). We observed a more intense Ki-67 staining to be present in poorly differentiated carcinoma, as shown in Figure 5 but we were unable to prove this statistically (Figure 4b). Similarly, no apparent relevance was observed in relation to surgical−pathological staging (Figure 4a).

Histodifferentiation distribution within different EC groups tended to be usual, as high differentiation grade cases were diagnosed most frequently (45.7%) and a poor differentiation grade was found in 17.2% of EC cases (Table 1).

Analysis of EC groups revealed that in EC cases associated with benign polyps, the epithelial Ki-67 score was higher than in EC, which developed some time after polyp resection (33.5 vs. 1.9%, *p* = 0.019). Ki-67 epithelial (33.5%) and stromal (1.08%) scores in cases of EC associated with benign polyps were higher compared with EC foci in polyps (epithelial 2.8%, stromal 0.18%, *p* = 0.000 and *p* = 0.008, respectively) but no apparent difference in clinicopathological characteristics was found between these EC groups.

## 4. Discussion

This study was designed to pursue evidence that the malignization potential of asymptomatic postmenopausal polyps is low and the result demonstrated low proliferative activity, not only in benign postmenopausal EPs but also in malignant ones. Even though EPs become malignant, their generally high histodifferentiation, conducive histology, low proliferation indices and early symptoms suggest a favourable prognosis.

Endometrial polyps are localized projections of endometrial tissue, more frequent among women in the Western world and they are usually considered benign endometrial lesions [16]; hence, their formation with or without atypical features might be an intermediate stage of the development of endometrial carcinoma. 

A systematic review and meta-analysis that involved a total of 10,572 patients who underwent polypectomy with histopathologic analysis showed an overall 3.57% malignancy of EPs. With respect to menopausal status, endometrial neoplasia was identified in 5.42% of women with EPs who were postmenopausal (relative risk (RR) 3.86; 95% confidence interval (CI) 2.92–5.11) compared with 1.70% of premenopausal women. With respect to abnormal uterine bleeding, 4.47% of women with symptomatic bleeding had malignant polyps (RR 3.36; 95% CI 1.45–7.80) compared with 1.51% of women without abnormal bleeding [5].

Current literature on the endometrial carcinogenesis process and biomarkers concerning the malignancy potential of EPs remains unclear and sparse. Previous immunohistochemical studies revealed that postmenopausal EPs appear to show variable intensity of oestrogen and progesterone receptor expression [17,18,19,20,21] and several studies have attempted to assess proliferation and apoptosis markers such as Ki67 [22,23], cyclin D1 [21], p53 [24,25] and bcl-2 [17] in the endometrial polyps, with different results. 

Uncontrolled proliferation is a hallmark of cancer [15]. Ki-67 pertains to non-histone proteins and is an affirmative proliferation marker at present. In the cell cycle, its expression begins to appear in phase G_1_, increases in phases S and G_2_, reaches the peak in phase M and disappears rapidly in the advanced stage of cell division but it is not expressed in phase G_0_ [26]. During the proliferative phase of the menstrual cycle, the expression of Ki-67 is normally enhanced [27]. The proliferation status of tumours is most widely measured using Ki-67, which becomes an important marker in clinical treatment decisions. An accurate estimation of the tumour proliferation is of high importance to exclude patients with slowly proliferating tumour cells and to avoid overtreatment [28]. Ki-67 becomes the most reliable index in the detection of tumour cell proliferation activity due to its short half-life period [26], thus the expression of Ki-67 reflects the tumour proliferation rate and correlates with initiation, progression, metastasis and prognosis of a number of types of tumours; for example, correlation between Ki-67 expression and patients’ survival is proven in cervical, uterine and breast cancers, non-Hodgkin’s lymphoma and large bowel cancer [29].

Our study demonstrated that the expression of Ki-67 is different among different endometrium lesions. Positive expression of Ki-67 was dramatically higher in EC and premenopausal EPs, indicating that epithelial Ki-67 percentage goes from a median expression of 2.41% in atrophic endometrium to a median of 4.67% in postmenopausal polyps, to the highest median expression of 11.4% in premenopausal polyps and 8.3% in EC, which was statistically significant between all groups except EC and premenopausal polyps. The highest median Ki-67 expression in premenopausal polyps may be explained by the physiological influence of estrogenic stimuli. The mean percentages of Ki-67 positive cells in carcinoma of the bladder, skin and colon are approximately 40% to 70%, whereas that of EC ranges from 20% to 30% [10,30,31,32,33]. These observations are consistent with our findings only in the group where EC was found with coexisting benign polyps (33.5%), whereas in other EC groups, Ki-67 antigenic activity was significantly lower, for example, 2.8% in EC focus in a polyp (*p* = 0.019) and 1.3% in EC cases with a previously resected polyp (*p* < 0.000). 

It was striking and interesting that all cases of EC in polyps commonly showed very low positive immunoreactivity, which was as low as that of benign tissue, such as benign asymptomatic EPs (*p* = 0.171) or atrophic endometrium (*p* = 0.112). Meanwhile, EC coexisting with benign EP manifested with a high and characteristic to EC proliferative activity, letting us speculate that benign EPs in latter cases were accidental findings. Moreover, the relatively long interval (in two cases, EC was diagnosed over 7 years after polyp resection) between the resection of the EP and EC diagnosis supports the speculation that EP may not be a precursor lesion to EC in our research but has naturally the same causes as EC (epidemiological association). Our findings support the idea that even though endometrial polyps become malignant, their low proliferative activity and conducive histology suggests a favourable prognosis.

We found a strong positive correlation in Ki-67 staining between the glandular and stromal compartments in all types of endometrium lesions, also showing statistically significant differences between the same groups. This may mean that the proliferative process mediated by oestrogen and, as indicated, by the Ki-67 marker occurs comparably in different tissue compartments of the polyp [34].

The Ki-67 expression is higher in EPs in postmenopausal women compared with those who have atrophic endometria [34]. In this study, we also observed a higher antigenic expression of Ki-67 in postmenopausal EPs than in atrophic endometria, which supports the hypothesis that the proliferation process is important in the pathogenesis of EPs. Oestrogen promotes proliferation and growth of the endometrial lining, while progesterone antagonizes oestrogen-driven growth and promotes differentiation [35]. Weakly proliferating glands are not uncommon after menopause and this is due to the response of the uterine mucosa to continuous low−level estrogenic stimulation. It is reported that postmenopausal women have an increased ability to convert androstendione, mainly of adrenal origin, into estrone in the adipose tissues of the body using the enzyme aromatase [36]. Our results show that the majority of postmenopausal endometria, despite ceased ovarian endocrine activity, retain a weak proliferative pattern for many years. Our study and other reports also showed that completely negative staining of Ki-67 was found only in atrophic endometrium (22.6% of cases were Ki-67 negative) [37,38].

Unlike other studies, with the increase in tumour differentiation grades, clinical stage and myometrial invasive depth, the expression rates of Ki-67 did not increase significantly, which might be related to the small number of included cases, group heterogeneity and atypical EC behaviour in the group of malignant polyps.

Despite the original and novel nature of our study, we acknowledge several limitations that restrict our ability to draw firmer conclusions. Firstly, this study was retrospective and therefore may be prone to selection and information biases. Secondly, as our study was confined to one hospital (nevertheless, a University Hospital), limited numbers of cases within histologic subgroups were identified, which caused our inability to substantiate some of the findings, not only regarding tendency but also regarding statistically significant level.

## 5. Conclusions

Our results indicate that Ki-67 has a role in the differentiation and categorization of endometrial lesions. Therefore, the validation, standardization concerning quantification and implication into clinical practice of Ki-67 as an endometrial biomarker may improve the process of defining patients who would benefit from a less aggressive treatment. 

Asymptomatic postmenopausal endometrial polyps exhibit low proliferative activity, suggesting low malignization potential and may not require resection in asymptomatic women. Further studies with larger samples and other biomarkers are needed to validate or refute these findings.

## Figures and Tables

**Figure 1 medicina-55-00543-f001:**
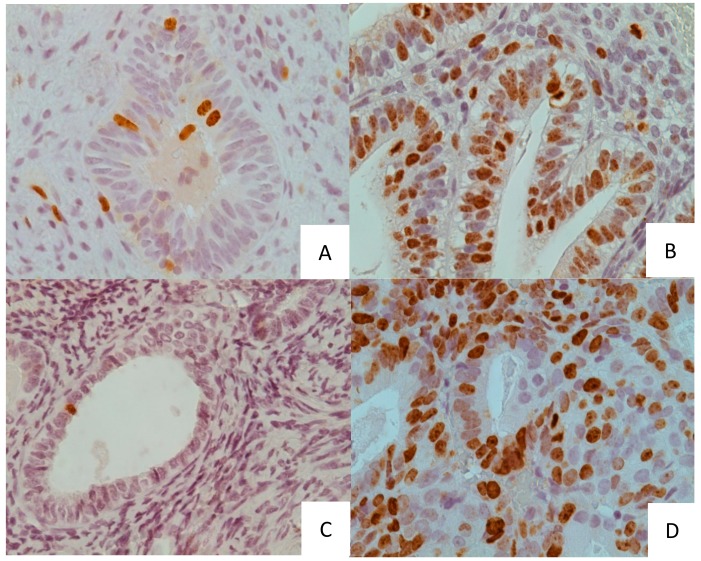
(**A**) A case of postmenopausal benign polyp. (**B**) A case of premenopausal benign polyp. (**C**) A case of atrophic endometrium. (**D**) A case of endometrial cancer. Original magnification 40× (**A**–**D**).

**Figure 2 medicina-55-00543-f002:**
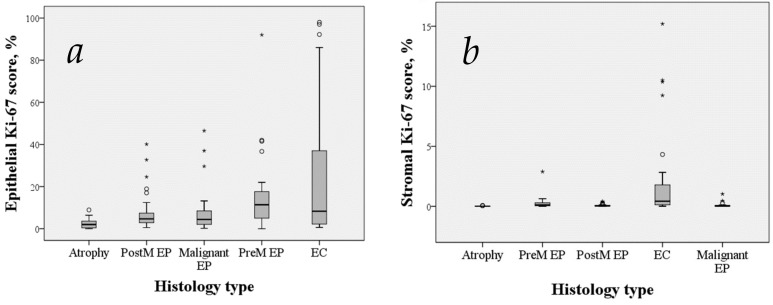
Box and whisker plots showing epithelial (**a**) and stromal (**b**) expression of Ki-67 over the spectrum of endometrial lesions. Boxes extend from the 25th to 75th percentiles and whiskers mark the range. Medians are displayed as horizontal lines within the boxes. The circles and asterisks represent points more than 1.5 IQR (interquartile range) and 3 IQR from the nearer quartile, respectively. Abbreviations: PostM EP, postmenopausal benign endometrial polyps. PremM EP, premenopausal benign endometrial polyps. EC, endometrial cancer. Malignant EP, endometrial polyps including polyp tissue of cases where polyp was found with coexisting endometrial cancer or had carcinoma focus inside.

**Figure 3 medicina-55-00543-f003:**
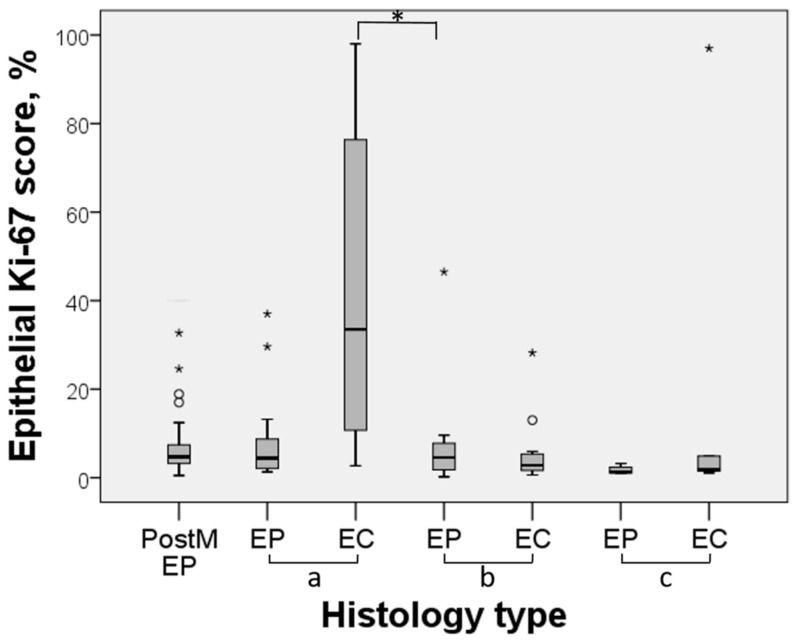
Box and whisker plot showing epithelial expression of Ki-67 over the spectrum of endometrial lesions in selected cases. Boxes extend from the 25th to 75th percentiles and whiskers mark the range. Medians are displayed as horizontal lines within the boxes. The circles and asterisks represent points more than 1.5 IQR and 3 IQR from the nearer quartile, respectively. *, *p* < 0.05. Abbreviations: PostM, postmenopausal polyps. EP, endometrial polyp. EC, endometrial carcinoma. a—cases of endometrial carcinoma with coexisting benign endometrial polyps; b—cases of endometrial polyps with foci of endometrial carcinoma; c—cases of endometrial polyps and later developed endometrial carcinoma.

**Figure 4 medicina-55-00543-f004:**
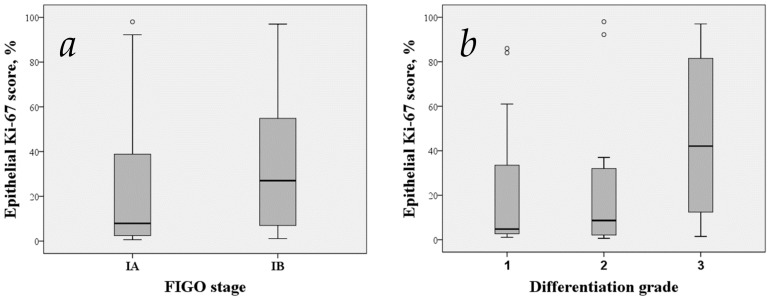
Comparison of Ki-67 scores according to staging (**a**) and histodifferentation (**b**). Boxes extend from 25th to 75th percentiles and whiskers mark the range. Medians are displayed as horizontal lines within the boxes. The circles are points more than 1.5 IQR from the nearer quartile. Abbreviations: FIGO, The International Federation of Gynaecology and Obstetrics.

**Figure 5 medicina-55-00543-f005:**
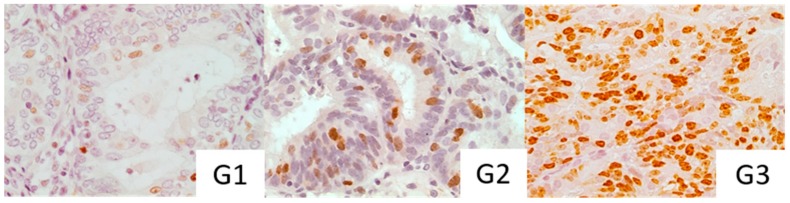
Immunohistochemical staining with Ki-67 in different grades of histodifferentation. G1: high differentiation degree. G2: moderate differentiation degree. G3: poor differentiation degree.

**Table 1 medicina-55-00543-t001:** Clinicopathologic and demographic characteristics of malignant endometrial polyps.

Characteristics	*n* (%)
Demographics	
Age	
<60	2 (16.7)
≥60	10 (83.3)
Vaginal bleeding	
No	4 (33.3)
Yes	8 (66.7)
Pathological classification	
Endometrioid adenocarcinoma	12 (100.0)
Surgical-pathological staging	
IA	12 (100.0)
IB	0 (0.0)
Differentiation degree	
High (G1)	7 (58.3)
Moderate (G2)	4 (33.3)
Poor (G3)	1 (8.4)
Depth of myometrial invasion	
Absent	7 (58.3)
<1/2	5 (41.7)
≥1/2	0 (0.0)
Endometrial carcinoma (EC) confined to a polyp	
Yes	7 (58.3)
No	5 (41.7)

**Table 2 medicina-55-00543-t002:** Ki-67 percentage in different endometrial lesions.

Histology Type	N	Ki-67			
		Median (IQR), %		*p*-Value * in Epithelium	*p*-Value * in Stroma
		Epithelial	Stromal		
Postmenopausal benign polyps	52	4.7 (3.2–7.8)	0.05 (0.02–0.08)	-	-
Premenopausal benign polyps	32	11.4 (5.0–17.6)	0.12 (0.05–0.30)	**0.003**	**0.002**
Endometrial cancer:	35	8.3 (2.2–37.0)	0.43 (0.13–1.79)	**<0.001**	**<0.001**
EC with coexisting benign polyps:	19				
Polyps		4.4 (2.1–8.8)	0.03 (0.01–0.06)	0.886	0.071
EC		33.5 (10.7–76.4)	1.08 (0.48–2.59)	**<0.001**	**<0.001**
Polyps with foci of EC:	12				
Polyp tissue		4.6 (1.8–7.8)	0.05 (0.02–0.11)	0.711	0.986
EC focus		2.8 (1.7–5.4)	0.18 (0.04–0.37)	0.171	**0.022**
Patients who had EP and later developed EC:	4				
Polyps	4	1.3 (1.1–2.4)	0.02 (0.02–0.04)	**0.015**	0.098
Later developed EC	4	1.9 (1.5–4.9)	0.28 (0.06–0.44)	0.242	**0.022**
Atrophic endometrium	31	2.4 (0.4–3.6)	0.01 (0.00–0.02)	**<0.001**	**<0.001**

* Compared with asymptomatic postmenopausal endometrial polyps (Mann Whitney *U* test). Abbreviations: EC, endometrial carcinoma; EP, endometrial polyps; IQR, interquartile range. Bold values denote statistical significance at the *p* < 0.05 level.

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
