# Peer review of "Proliferation in Postmenopausal Endometrial Polyps—A Potential for Malignant Transformation"

_medicina, 2019, doi:10.3390/medicina55090543_

Round 1

Reviewer 1 Report

In this retrospective study, the authors investigated the proliferation within endometrial polyps as one of the indicators of their oncogenic potential in asymptomatic postmenopausal women and conclude that benign postmenopausal endometrial polyps exhibit low proliferative activity suggesting low malignant potential.

The use of “low malignant potential” is misleading, since tumors of “low malignant potential” are usually defined by pathological findings. For example, the endometrioid ovarian tumor of low malignant potential applies to a neoplasm composed predominantly of the cytologically malignant epithelium of endometrioid type without destructive stromal invasion. Thus I do not understand that postmenopausal benign endometrial polyps without cytological atypia may be “low malignant potential.” 

In “Conclusions”, the authors state, “Ki-67 has a role in endometrial carcinogenesis pathways.” However, in this study, no cases of atypical endometrial hyperplasia, which is usually the premalignant lesion of endometrioid carcinoma, were included. In addition, in polyps with foci of endometrial carcinoma (EC), Ki-67 percentage was lower in EC focus than in polyp tissue. Thus I don’t think the authors can discuss the role of Ki-67 in endometrial (endometrioid) carcinogenesis from the results of this study.

Also, in “Conclusions”, the authors state, “Ki-67 may optimize the treatment of asymptomatic postmenopausal endometrial polyps.” I agree with this, but how do we know the expression of Ki-67 before polypectomy?

Use the same term consistently for a specific object: endometrium cancer in L31 and endometrial carcinoma in L32.

Abbreviation usage. EP, which is defined in Line 42, is often not used thereafter in the manuscript.  

Reviewer 2 Report

The authors present an original study on the proliferation in postmenopausal endometrial polyps as measured by the Ki-67 score. They found that Ki-67 has a role in endometrial carcinogenesis pathways and may perform as a potential biomarker to categorize endometrial lesions. The authors also conclude that by the use of Ki-67 the treatment of asymptomatic postmenopausal polyps may be optimized.

This is an interesting study, an important subject as I think we should be cautious about treating endometrial polyps in asymptomatic women. Because in many countries this is still very common and women are potentially being overtreated, studies on this subject are very important.

I have some general comments:

As this study aimed to investigate the proliferative activity in postmenopausal endometrial polyps, I don’t understand why the authors used quite a significant part of the results section to describe the proliferative activity in the different grades and stages of endometrial carcinoma. As concluded in the discussion by the authors, this study is too small to draw conclusions on this and this was not the research question, so why mention this? It makes the article somewhat confusing to read.

Secondly, as this is quite a small study I wonder what the results of the sample size calculation were? Sample size is shortly mentioned in the statistical analysis section, but which difference was considered significant? what was the hypothesis? and what is the result of the sample size calculation? how many cases were needed to show a significant difference in proliferative activity? Please explain more in detail.

            For this study the authors choose to compare proliferative activity in benign endometrial polyps versus endometrial carcinoma. I think it would also be very interesting to compare proliferative activity in symptomatic versus asymptomatic endometrial polyps.

Some more specific comments:

Line 60: what is the evidence to use Ki-67 as a marker for EC? please explain more and give references. The authors give references in the methods and discussion section, but I think some more explanation would be helpful in the introduction.

Line 62: activities should be ‘activity’ 

Line 72-84: I took me a few times reading this piece to understand how and why the authors grouped the cases of endometrial carcinoma and endometrial polyps. Maybe explain more clear.  Same comment for the results in Table 2. 

Line 75: why did you also examine 4 cases of benign endometrial polyps who later developed cancer? What is the added value?

Line 77: ‘control’ should be ‘controls’ and the comma should be replaced by ‘and’ for better reading and understanding

Line 78: how was menopause defined? did you mean: Postmenopausal women were defined as women with at least 12 months amenorrhea? please rephrase

Line 82: ‘The histologic type…’ please remove and transfer to results section.

Line 116-118: as stated above: please explain more about the sample size calculation.

Line 132: was this number based on the sample size calculation? Or did you just have 150 cases, as this is a retrospective study?

Table 1: clinical characteristics of endometrial polyps would have been interesting as well. Age and bleeding pattern?

Table 2: What is the difference between cases ‘polyps which later developed EC’ and ‘EC which previously had polyp’?

Line 168-169: how do you explain this result?

Line 206-208: the fact that also malignant polyps show low proliferative activity is an interesting finding and the suggestion that this suggest a favorable prognosis is well worthy to study more. Please give this some more attention in the discussion. There is also some evidence available that EC is a coincident finding in the pathology results of uteri removed for other reasons. 

Line 209: in intro you mention 8% in general population and 20% in pmp women. why mention different percentage here with different reference? This is a  bit confusing reading the article..

Line 217-219: Difference between symptomatic and asymptomatic postmenopausal women (4.47% malignant polyps versus 1.51%) is much more relevant for the aim of this article? why mention different percentages here than in introduction?

Line 226-238: please refer to evidence on the use of Ki-67 in women with endometrial carcinoma specifically.

Line 266-268: please rephrase.

Line 268: our results show instead of support?

Line 274: small amount of study group? do you mean small amount of included cases?

Line 278: ‘confined to the one hospital’ please remove ‘the’ 

Round 2

Reviewer 1 Report

The legend of Figure 4. Panel a is on FIGO stage, and Panel b is on histodifferentiaion.

The legend of Figure 5. G1: good differentiation degree. However, in Table 1, differentiation degree is classified into high, moderate, and poor. Use the same term consistently.
